# The Prevalence of Lung Carcinoma Among the Population in Bulgaria and Its Association with Radon Exposure in Residential Buildings

**DOI:** 10.3390/ijerph22121841

**Published:** 2025-12-09

**Authors:** Evgenia Todorova

**Affiliations:** Department of Radiotherapy and Metabolic Brachytherapy, University Hospital “Queen Giovanna–ISUL”, Medical University of Sofia, 1431 Sofia, Bulgaria; etodorova686@gmail.com

**Keywords:** lung carcinoma, incidence, prevalence, radon exposure in dwellings, environmental epidemiology, public health surveillance

## Abstract

**Highlights:**

**Public health relevance—How does this work relate to a public health issue?**
Indoor radon is a major environmental carcinogen and the second leading cause of lung carcinoma after smoking.Bulgaria shows notable regional differences in lung cancer incidence, which correspond to variations in indoor radon levels.

**Public health significance—Why is this work of significance to public health?**
The study demonstrates moderate to very strong correlations between indoor radon indicators and regional lung carcinoma incidence.The results support using the WHO-recommended reference level of 200 Bq/m^3^ in national radiation-protection policies.

**Public health implications—Key implications for practitioners, policymakers, and/or researchers**
High-radon regions should be prioritized for mitigation programs, public awareness, and improved screening pathways.Provides national evidence to guide radon monitoring programs, updates to building regulations, and broader public health prevention efforts.

**Abstract:**

Background: Lung carcinoma remains the leading cause of cancer-related mortality worldwide, with smoking as the primary risk factor and radon exposure as the second, and the first among non-smokers. The combined effect of tobacco smoke and indoor radon increases disease risk up to 2.5 times, emphasizing the need for prevention and environmental risk assessment. Methods: This study analyzed the incidence of lung carcinoma in Bulgaria during 2013–2022 and examined its association with indoor radon exposure across different regions. Annual data were obtained from the National Statistical Institute and the National Survey of Indoor Radon Concentrations in Residential Buildings (2015–2016). Results: The average annual incidence was 43.5 per 100,000 population, showing a 3.4% annual decline, while the average prevalence was 131.7 per 100,000, decreasing by 1.4% per year. Considerable interregional variation was observed, with incidence ranging from 25.5 to 62.4 per 100,000. A moderate positive correlation was found between lung carcinoma incidence and mean indoor radon concentration, and a stronger to very strong correlation with the proportion of dwellings exceeding 300 Bq/m^3^ and 200 Bq/m^3^. Conclusions: These findings indicate a positive association between residential radon exposure and lung cancer morbidity and support maintaining the WHO-recommended reference level of 200 Bq/m^3^.

## 1. Introduction

According to the latest data from the International Agency for Research on Cancer (IARC) [1], in 2022, 20 million people worldwide developed malignant diseases. Of these, 2.5 million cases were lung carcinoma, which, with a share of 8%, places lung carcinoma as the leading malignant disease globally. Similar values have been reported in previous years.

Worldwide, regardless of national and regional differences in distribution, lung carcinoma remains the leading cause of death from malignant diseases [1]. The incidence of lung carcinoma varies geographically, reflecting the differences in smoking prevalence, which is considered the main risk factor. In developing countries, with economic growth, the number of smokers also increases. The peak rise in smoking prevalence usually occurs earlier among men and subsequently among women, with the corresponding increase in lung cancer incidence emerging in the following decade [1,2,3].

In continental Europe, there are significant geographical differences in lung cancer incidence. The highest rates are observed in Central and Eastern Europe—among men, incidence reaches 76.6 per 100,000 in Hungary, 71.3 in Serbia, and 61.1 in North Macedonia. Since 1990, a sustained downward trend has been observed, explained by reductions in smoking [4].

According to data from the National Statistical Institute (NSI) [5], in Bulgaria in 2022, there were 26,235 registered cases of malignant diseases. The leading site was breast cancer (13.5%), followed by cancers of the respiratory system (10.8%), of which 8.5% were lung carcinoma (trachea, bronchi, and lung parenchyma). Deaths from malignant neoplasms numbered 17,221, with cancers of the respiratory system ranking first, accounting for 21% of all cancer-related deaths.

Radon is considered the second leading cause of lung carcinoma in the population, and the leading cause among non-smokers. It is a major component of natural background radiation, contributing 50% of internal exposure from natural sources. Radon (Rn-222) is a radioactive gas, a decay product of uranium (U-238), with a half-life of 3.8 days, allowing it to diffuse from the soil before decaying into alpha particles and a series of short-lived radioactive elements (Po-218, Po-214), down to Pb-210, a solid particle with a half-life of 22 years. Radon progeny accumulate in enclosed spaces, attach to household dust particles, and once inhaled, become a cumulative source of internal irradiation. The inability of radon to disperse in indoor environments leads to significant increases in volumetric concentrations, further augmented by radon migrating from building materials.

The established consensus that high radon exposure is a risk factor for lung carcinoma originates from numerous studies among non-smoking miners in silver and uranium mines [6,7,8]. Lubin and Boice [9], in a meta-analysis of 11 cohort studies (65,000 miners and 2700 deaths), demonstrated a strong linear association between lung cancer risk and cumulative radon exposure. This indicated that even very low levels—such as those present in homes—pose a substantial risk for lung cancer development. The authors also found that risk is greater when the cumulative dose is received over a longer time period. Following these fundamental studies and the recognition by the WHO of indoor radon as a Group 1 human carcinogen [10], the issue of indoor radon exposure has become particularly relevant, both in terms of health risk and prevention opportunities.

In Bulgaria, the reference levels for indoor radon concentration are defined in the Regulation on Basic Norms for Radiation Protection issued in 2012 and revised in 2018, establishing a reference level of 300 Bq/m^3^ for existing buildings and 200 Bq/m^3^ for new constructions. Council Directive 2013/59/Euratom recommends a reference level of 300 Bq/m^3^, but emphasizes that a statistically significant increase in lung cancer risk is observed at concentrations as low as 100 Bq/m^3^. This evidence should be taken into account when developing and updating national radiation protection standards [11,12,13].

Multiple meta-analyses confirm the dose–response relationship between indoor radon exposure and lung carcinoma. In Europe, Darby et al. (2005) conducted a representative meta-analysis of 13 case–control studies, establishing [14] a linear increase in relative risk (RR) of 16% per 100 Bq/m^3^ of indoor radon. This positive linear relationship was observed both among smokers and non-smokers [15,16].

Further insights into disease etiology and prevention strategies have been provided by detailed studies of the combined effect of radon and tobacco smoke. Initially considered synergistic, this interaction is now recognized as more than additive. In a large Swedish case–control study, Lagarde et al. [17] found that, among non-smokers, radon exposure in homes was significantly more harmful for those also exposed to environmental tobacco smoke. Similar results were reported in a large-scale South Korean study [18], which confirmed significant effects of both radon and tobacco smoke—individually and combined—on lung carcinoma risk. Even at concentrations below established regulatory limits, radon can increase risk more than 2.0–2.5 times [17,18,19].

This combined effect stems from the ability of radon progeny to attach to dust particles contained in both the mainstream and side-stream smoke of cigarettes. A smoker exposed to radon has a much greater risk of developing lung carcinoma [20]. For passive smokers living with smokers, the risk is 20–30% higher [21]. Increased potential alpha activity has also been observed with electronic cigarettes, due to radon progeny attaching to the higher number of aerosol particles [22]. The aim of the present study is to investigate the distribution of lung carcinoma over a ten-year period in Bulgaria and, based on the average annual incidence in the respective period across individual regions, to analyze the association between the disease and radon exposure in residential dwellings.

## 2. Materials and Methods

The epidemiological indicators for the distribution of lung carcinoma—incidence and prevalence—were calculated based on annual data of registered and newly diagnosed cases published by the National Statistical Institute (NSI) [5]. For the period 2013–2022, the analysis included: the dynamics of distribution in the country during the observation period; the average annual incidence and prevalence for the study period nationwide and separately for all regions. The following indicators were calculated for the evaluation: incidence rate, prevalence rate, and the average duration of the disease.

**Prevalence** is an indicator of the portion of the population that suffers from a given disease within a specific period of time. It is influenced by two main factors: the frequency of new cases, which increases the number of patients, and the frequency of deaths or recoveries among the sick during the same period, which decreases it. Based on annual data of registered patients with lung carcinoma, the average annual prevalence rate was calculated for the period 2013–2021 for the country and all 28 regions.

**Incidence rate** is the index of newly diagnosed cases during a specific period. The average annual incidence rate was calculated based on newly registered cases each year in the period 2013–2022, for the country and all 28 regions.

**Average duration of the disease** was calculated using the mathematical relationship between prevalence (*P*) and incidence (*I*), expressed as [22]:*P = I × average disease duration*
or*Average disease duration = P/I*

The result represents the average length of disease duration, including both fatal and recovered cases.

The results obtained were analyzed with respect to:-the dynamics of average annual prevalence and incidence in the country during 2013–2022, calculated as the difference from the previous year;-the dynamics of average annual disease duration during the observation period, calculated as the difference from the previous year.

Radon exposure data were obtained from the published “National Survey of Indoor Radon Concentration in Residential Buildings, 2015–2016” [23].

The national survey was conducted over a one-year period in two phases (summer and winter), which together were used to determine the annual average indoor radon concentration in residential buildings. Radon concentration was measured in 2776 dwellings, selected proportionally to the population distribution across all 28 administrative regions of the country. For each region, the annual average concentration was derived from measurements taken in 88 to 106 dwellings. The survey reported the following results:-The national annual average indoor radon concentration, based on the sampled dwellings, was 111.2 Bq/m^3^.-The distribution of indoor radon concentrations followed a log-normal pattern, with values ranging from 12 to 1314 Bq/m^3^.-An estimated 4.4% of the population resides in dwellings with radon concentrations above the regulatory reference level of 300 Bq/m^3^.-Accordingly, 95.6% of dwellings were below the 300 Bq/m^3^ reference level.-The proportion of dwellings with indoor radon concentrations exceeding the WHO-recommended level of 200 Bq/m^3^ was 12%.

In the present study, the following indicators from the national survey were used: the annual average radon concentration by region, and the expected percentage of dwellings with indoor radon concentrations above 300 Bq/m^3^ and above 200 Bq/m^3^.

Although the national radon survey was conducted in 2015–2016, numerous studies have shown that indoor radon concentrations exhibit high temporal stability because they are determined primarily by stable geological conditions and long-lasting building characteristics. Consequently, radon measurements from a single national survey are considered suitable for long-term epidemiological analyses at the population level.

To investigate the relationship between lung carcinoma distribution and indoor radon exposure, Spearman’s rank correlation analysis was applied. Statistical analyses were performed using Microsoft Excel (Microsoft Corporation, Redmond, WA, USA). Spearman’s rank correlation coefficients (two-tailed) were calculated to assess the relationship between regional lung carcinoma incidence and indoor radon indicators.

To minimize the influence of confounding factors on radon exposure duration in the population, population movement between regions was assessed on the basis of net migration (mechanical growth) [24]. For statistical analysis, only regions with zero and/or negative net migration rates during the entire period (2015–2023) were included. The excluded regions were: Sofia City, Burgas, Varna, Plovdiv, Kardzhali, Stara Zagora, and Pernik. The population in the included regions has a higher age-dependency ratio compared with the national average (61.0% in 2023), meaning fewer than two working-age persons for each dependent individual.

Occupational radon exposure was excluded as a confounding factor, since all uranium mines in the country, whether depleted or undepleted, were closed after 1990 [25].

## 3. Results

The data on the distribution of lung carcinoma in the country show that the incidence for the period 2013–2022 was 43.5 cases per 100,000 population (Table 1).

The average annual prevalence for the observed period was 131.7 cases per 100,000, which is about three times higher than the incidence.

During the observation period, both frequency indicators demonstrated a general downward trend (Table 1, Figure 1). The only exception was in 2019, when the incidence rate compared to the previous year increased by 13.5%, and the prevalence increased by 4.4%. These peaks were followed by a sharp decline.

Hypothetically, these fluctuations in frequency may be associated with increased public attention to respiratory symptoms triggered by widespread media coverage of the emerging pandemic. This heightened awareness may have prompted more frequent medical consultations, effectively functioning as a form of incidental screening. Conversely, during the peak of the COVID-19 pandemic in 2020–2021, access to and utilization of outpatient medical services were markedly reduced, which likely contributed to a subsequent decline in diagnosed cases. After 2019, the declining trend of both indicators continued.

The regional frequency of lung carcinoma distribution is presented in Figure 2. The data clearly demonstrate substantial geographical variation in the average annual incidence of lung carcinoma across the country. Between regions, incidence varied from 25.5 cases per 100,000 in Sofia Region to 62.4 cases per 100,000 in Pleven Region. Rates were lowest in the western parts of the country and increased toward the northeast and southeast.

Data from the National Survey of Indoor Radon Concentrations in Residential Buildings (2015–2016) [23] are presented and discussed in accordance with the reference levels in the Regulation on Basic Norms for Radiation Protection [11], specifically:-“Average annual radon concentration” in Bq/m^3^,-“Percentage of expected dwellings (log-normal distribution) with radon concentrations above 300 Bq/m^3^,” and-According to WHO recommendations [10], the “Expected percentage of dwellings above 200 Bq/m^3^.”

The survey results showed:-An average annual radon concentration of 111.2 Bq/m^3^;-Significant interregional variation in average annual values—from 76.0 Bq/m^3^ in Montana Region to 212.2 Bq/m^3^ in Yambol Region;

Considerable regional differences in the expected percentage of dwellings exceeding 300 Bq/m^3^ (regulatory reference level) and 200 Bq/m^3^ (WHO recommended level).

The association between lung carcinoma incidence and indoor radon exposure is presented in Table 2. The analysis was based on data from 21 regions for the period 2012–2023.

The results in Table 2 show a moderate linear correlation (r = 0.42) between lung carcinoma incidence among the population of different regions and the average radon concentration in dwellings.

A high degree of correlation (r = 0.89) was found between incidence and the percentage of dwellings exceeding 300 Bq/m^3^.

A very high degree of correlation (r = 0.95) was observed between incidence and the percentage of dwellings exceeding 200 Bq/m^3^.

## 4. Discussion

The data on the distribution of lung carcinoma in Bulgaria show that the average annual incidence during the period 2013–2022 was 43.5 per 100,000 population. This value is close to the most recent figures for Greece (40.5 per 100,000) and Serbia (47.31 per 100,000) [11]. Among European countries, the highest incidence is reported in Hungary (50.1), and the lowest in Belgium (28.1) [26].

During the study period, the incidence rate decreased by 3.4% annually, which can be explained by changes in health-prevention culture in the population and by sustained policies limiting smoking in workplaces, public buildings, and restaurants. Similar results were reported by Bryant-Genevier et al. in the United States, where lung cancer incidence among men declined by approximately 2.6% annually during 2010–2020, with significant regional and racial differences in the rate of decline [27].

The trend of decreasing prevalence was less pronounced. This is due to the fact that the average duration of the disease steadily increased during the observation period, from 2.68 years to 3.38 years. Survival among patients with lung carcinoma is the shortest of all malignant diseases. While the 5-year survival rate in Europe is about 15%, in Bulgaria it is only 8%, primarily due to late diagnosis [28]. The observed increase in average disease duration in 2013–2021 may be interpreted as an indicator of improvements in diagnosis during this ten-year period.

The average annual incidence of lung carcinoma also showed considerable regional differences, ranging from 25.5 cases in the Sofia Region to 59.6 and 62.4 in Haskovo and Pleven Regions, respectively.

These regional differences in frequency may be related to factors with a well-established causal link to the development of lung carcinoma and which play a determining role in its epidemiological characteristics. Foremost among these is radon exposure—both in residential and occupational settings. Occupational radon exposure was excluded as a potential confounding factor, as all uranium mines in the country were closed after 1990 [25]. We consider that the observed regional discrepancies are primarily associated with the geological structure of the subsurface layers and, consequently, with the resulting indoor radon concentrations in dwellings. Tobacco smoke is a co-factor that increases the effective radon dose indoors; however, no national data are available regarding regional variations in smoking behaviors or in the prevalence of smoking across the population.

The relationship between radon exposure and lung carcinoma in the population has been extensively studied in case–control designs, where radon measurement devices are placed in the homes of patients and controls. Detailed findings have been obtained in meta-analyses of such studies [14,26,29]. The most frequently cited results are those of Darby et al. [14], which demonstrated a significant dose–response relationship between lung carcinoma and radon levels.

The results of the present study show a moderate linear correlation (r = 0.42) between lung carcinoma incidence in the population of different regions and the average indoor radon concentration. This is noteworthy, since the mean annual radon levels across regions ranged only from 76.0 to 212.2 Bq/m^3^, and some regions include additional ethnic variations.

A high degree of linear correlation (r = 0.89) was established between incidence and the percentage of dwellings exceeding 300 Bq/m^3^, and a very high correlation (r = 0.95) between incidence and the percentage of dwellings exceeding 200 Bq/m^3^. These high correlation coefficients can be explained by the log-normal distribution of radon values underlying regional averages. The stronger correlation at the 200 Bq/m^3^ threshold indicates that even a relatively small number of dwellings in the 200–300 Bq/m^3^ range can contribute to lung carcinoma development, supporting the WHO’s recommended reference level of 200 Bq/m^3^.

No association was found between lung carcinoma incidence and the proportion of dwellings with indoor radon concentrations below 200 Bq/m^3^.

The results of the present study demonstrate a clear regional correlation between indoor radon exposure and the incidence of lung carcinoma in Bulgaria. However, these findings should be interpreted as indicative rather than confirmatory. Given the ecological design of the analysis and the absence of individual-level exposure data, the observed relationship reflects association rather than causation. Further analytical research—particularly case–control or cohort studies integrating smoking prevalence, demographic, and socioeconomic factors—is necessary to verify and quantify the causal contribution of radon exposure to lung cancer risk.

Similar ecological associations between indoor radon concentrations and lung cancer incidence have been observed in other European studies, particularly in regions with comparable geological characteristics and housing structures. These parallels reinforce the need for continuous monitoring of indoor radon levels and public awareness of the associated health risks.

This study has several limitations. It is based on aggregated regional data and on a single national radon survey conducted in 2015–2016, which may not fully capture local temporal variability. Potential confounders such as smoking habits, occupational exposures, and air pollution were not controlled for. Additional interregional factors may also influence the frequency of lung carcinoma. Regions with sizeable ethnic groups may exhibit distinct lifestyles and health-related behaviors, including different patterns of cigarette consumption; however, no national data are currently available on regional variations in smoking prevalence. Differences in population age structure represent another possible source of variation. This influence is partially mitigated in the present analysis by including only regions with zero and/or negative net migration throughout the period (2015–2023), which have a higher age-dependency ratio than the national average. Despite these limitations, the analysis provides valuable national-level evidence supporting the importance of indoor radon exposure as an environmental determinant of lung carcinoma.

## 5. Conclusions

The average annual incidence of lung carcinoma in Bulgaria during the ten-year period (2013–2022) was 43.5 per 100,000 population, and the average annual prevalence was 131.7 per 100,000. These values are comparable to those reported in neighboring countries. Throughout most of the observation period, both indicators demonstrated a stable downward trend, with the exception of the years 2019–2021. This temporary interruption is likely attributable to pandemic-related factors, including increased medical consultations in 2019 and reduced access to routine diagnostic services during the height of COVID-19 restrictions in 2020–2021.

Marked regional disparities in incidence were identified, ranging from 25.5 to 62.4 per 100,000 population. A moderate correlation was observed between lung carcinoma incidence and average indoor radon concentration, whereas a substantially stronger—up to very strong—correlation was noted with the proportion of dwellings exceeding 300 Bq/m^3^ and 200 Bq/m^3^. These findings underscore the importance of residential radon as an environmental determinant of lung cancer risk and further support the WHO-recommended reference level of 200 Bq/m^3^.

No changes to national radon reference levels occurred during the study period, meaning the regulatory framework remained consistent throughout, and therefore, no impact on measurements or data interpretation is expected. Both the 2012 and 2018 national regulatory frameworks preserved the same reference level of 300 Bq/m^3^ for existing buildings; therefore, no regulatory revisions during the study period could have influenced measurement interpretation or altered population exposure patterns.

Although the study is limited by its ecological design, the use of aggregated regional data, and reliance on a single national radon survey, the results provide meaningful national-level evidence of an association between indoor radon exposure and lung carcinoma. Future research should incorporate individual-level data and additional variables—such as smoking prevalence, geological characteristics, housing conditions, and socioeconomic factors—to more precisely quantify radon-related risk and strengthen the scientific basis for targeted prevention strategies.

## Figures and Tables

**Figure 1 ijerph-22-01841-f001:**
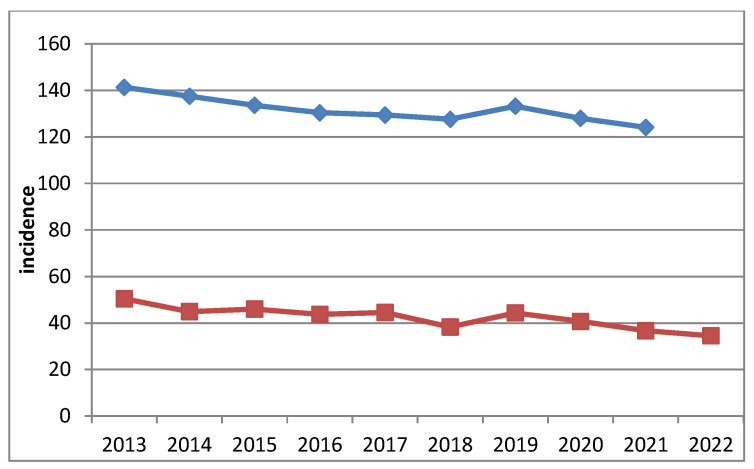
Dynamics of prevalence (blue) and incidence (red) of lung carcinoma in Bulgaria. The *Y*-axis represents the incidence per 100,000 population, and the *X*-axis represents the calendar year (2013–2022).

**Figure 2 ijerph-22-01841-f002:**
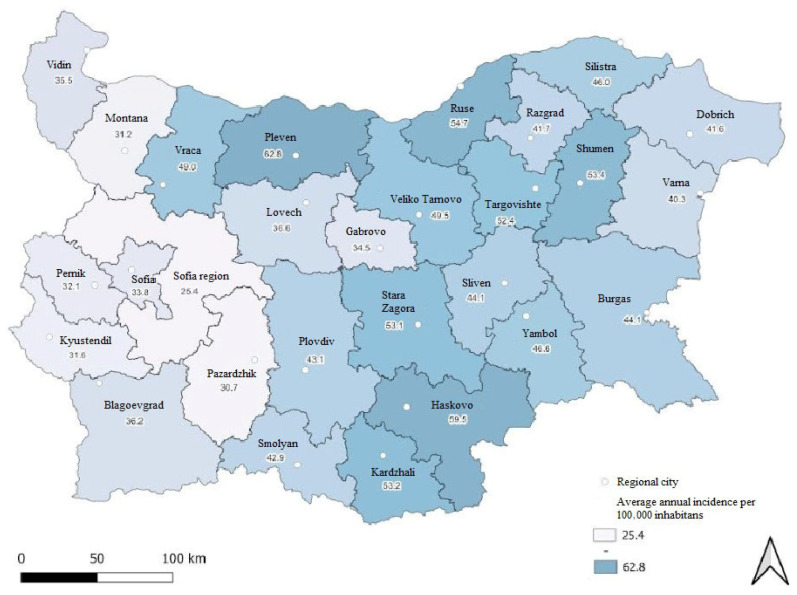
Average annual incidence of lung carcinoma across Bulgarian regions, 2013–2022.

**Table 1 ijerph-22-01841-t001:** Distribution of lung carcinoma in Bulgaria, 2013–2022 (per 100,000 population).

Year	Incidence Rate		Prevalence Rate		Average Duration of Disease in Years
	Cases per Year	% Change from Previous Year	Prevalence per Year	% Change from Previous Year	
2013	52.4		141.3		2.68
2014	49.9	−2.8	137.5	−2.7	2.75
2015	45.0	−5.0	133.6	−2.8	2.96
2016	43.7	−2.9	130.4	−2.4	2.98
2017	40.5	−7.3	129.4	−0.8	3.19
2018	38.3	−5.7	127.6	−1.4	3.33
2019	44.3	+13.5	133.2	+4.4	3.00
2020	40.7	−8.1	128.0	−3.1	3.14
2021	36.7	−9.8	124.1	−3.0	3.38
2022	34.5	−5.9	*		
Average annual rate	43.5	−3.4%	131.7	−1.4	3.09

* Note: National data for 2022 are presented differently in the official source.

**Table 2 ijerph-22-01841-t002:** Correlation between incidence of lung carcinoma and indicators of indoor radon exposure.

Variable Relationship	Range of Variation	Spearman’s rank Correlation Coefficient (rₛ)
Incidence rate of lung carcinoma (by region)/Mean indoor radon concentration (Bq/m^3^)	from 76 Bq/m^3^ to 212 Bq/m^3^	0.42
Incidence rate of lung carcinoma/Percentage of dwellings with radon concentration > 300 Bq/m^3^	from 08% to 17.5%	0.89
Incidence rate of lung carcinoma/Percentage of dwellings with radon concentration > 200 Bq/m^3^	from 4.4% to 43.6%	0.95
Incidence rate of lung carcinoma/Percentage of dwellings with radon concentration < 200 Bq/m^3^	from 38.9% to 95.5%	0.04

Note: All correlations were calculated using Spearman’s rank correlation coefficient (two-tailed test). All associations were statistically significant (*p* < 0.05).

## Data Availability

The data presented in this study are derived from publicly available sources. Lung carcinoma incidence data were obtained from the National Statistical Institute (NSI) of Bulgaria (https://www.nsi.bg, accessed on 4 April 2025). Indoor radon concentration data were obtained from the National Radon Program, Ministry of Health, Bulgaria. All datasets analyzed during the current study are available from these public repositories.

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
