# Peer review of "The Prevalence of Lung Carcinoma Among the Population in Bulgaria and Its Association with Radon Exposure in Residential Buildings"

_ijerph, 2025, doi:10.3390/ijerph22121841_

Round 1

Reviewer 1 Report

Comments and Suggestions for Authors

The paper analyzes the Prevalence of Lung Carcinoma Among the Population in Bulgaria and its Association with Radon Exposure in Residential Buildings.

The manuscript is clear, easy to read and quite interesting. However I have some suggestions to the author for improving the work before publishing.

Please remove from the abstract the words in bold.

Please, cite in the introdution section the last European Directive (59/2013/Euratom) and the national regulation.

Page 3 line 139 - the sentence is missing of a part

page 4 line 160 - covid pandemic started in the late 2019 in China and officially on March 2020. Please explain better the motivation because the statement and justification seems not to be correct.

Figure 2 and line 160-173 Please add a discussion in the subsequent section if this regional variation can be correlated to geology and so different mean of indoor radon concentrations or occupational and social factors.

Please extend the conclusion section with limitations, anomalies registered (2019) and other possible correlations or statistical studies. The data refers to a period from 2013 to 2022. Highly probably the National regulation was not the same since the repealing of the ED 59 /2013 was mandatory within 2018. In the passage from a regulation to another one was the national threshold value the same? Did change something in the regulation framework that could have affected data?

Author Response

Comments 1: Please remove from the abstract the words in bold.

Response 1: Thank you. The bold phrases were removed.

“Abstract, lines 9,14,17,23,27”

Comments 2: Please cite in the introduction section the last European Directive (59/2013/Euratom) and the national regulation.

Response 2: I have accordingly added and cited in the Introduction.

This change can be found – page number 2, lines 73-79, Citations added: references #24, #25.]

“In Bulgaria, the reference levels for indoor radon concentration are defined in the Regulation on Basic Norms for Radiation Protection (2012, amended 2018). The European Council Directive 2013/59/Euratom also establishes a reference level of 300 Bq/m³…”

Comments 3: Page 3 line 139 – the sentence is missing of a part.

Response 3: Thank you for pointing it out. This was due to an inadvertent formatting error during text transfer. I has since been removed.

Comments 4: Page 4 line 160 – COVID pandemic started in late 2019… the motivation seems not correct.

Response 4: Thank you for the observation. The explanation was revised to provide an accurate timeline and more precise justification. The changes made can be found – page number 6, lines 201-207.]

Hypothetically, these fluctuations in frequency may be associated with increased public attention to respiratory…

Comments 5: Figure 2 and lines 160–173: Please add discussion if regional variation correlates with geology, radon, occupational and social factors

Response 5: I agree and acknowledge that its inclusion in the primary manuscript would have been appropriate. A new discussion paragraph was added addressing geological influences, occupational factors, and social determinants. In the revised manuscript this change can be found – page number 9 and lines 277-286.]

These regional differences in frequency may be related to factors with a well-established…

Comments 6: Please extend the conclusion section with limitations, anomalies registered (2019) and other possible correlations or statistical studies. The data refers to a period from 2013 to 2022. Highly probably the National regulation was not the same since the repealing of the ED 59 /2013 was mandatory within 2018. In the passage from a regulation to another one was the national threshold value the same? Did change something in the regulation framework that could have affected data?

Response 6: I appreciate your insightful question, which has contributed to improving the quality of my work. The conclusion was expanded accordingly. Paragraph on regulatory stability were added. A new clarification was added addressing regulatory consistency. In the revised manuscript this change can be found under “Conclusion” – page number 10, and lines 341-368.]

“[he average annual incidence of lung carcinoma in Bulgaria during the ten-year period (2013–2022) was…

Reviewer 2 Report

Comments and Suggestions for Authors

The manuscript analyzes the annual average trends of lung carcinoma in Bulgaria between 2013 and 2022, along with a correlation analysis involving radon. The following points may help further improvement the manuscript:

  1. My primary concern relates to the radon concentration data, which were sourced from the “National Survey of Indoor Radon Concentration in Residential Buildings, 2015–” A key question is whether residential radon levels remain relatively stable over time. If so, the authors should provided justification and supporting evidence regarding the temporal stability of these concentrations. This would strengthen the validity of the correlation analysis conducted.
  2. The introduction could be better organized to more clearly articulate the background, rationale, and significance of the study.
  3. Table 1 and Figure 1 appear to present the same content. It is suggested to retain only one of them. Additionally, the Y-axis in Figure 1 should be clearly labeled, and the legend (though mentioned in the caption) should in included directly in the figure for better readability.
  4. The discussion section should also address other potential factors influencing lung carcinoma incidence. This would help contextualize the specific role of radon and highlight its significance relative toe other risk factors. If a detailed discussion of these factors is beyond the scope of the current manuscript, the authors should at least acknowledge these potential confounders in the conclusion or the study outlook to provide a more balanced perspective.
  5. The text in Lines 198–204, which currently appears as a single-sentence paragraph, should be consolidated into a coherent paragraph. Moreover, the conclusion section should be restructured—either presented as a continuous paragraph or as a numbered list (e.g., (1), (2), (3)...) for better clarity and organization.

Author Response

Comments 1: My primary concern relates to the radon concentration data, which were sourced from the National Survey of Indoor Radon Concentration in Residential Buildings, 2015–2016. A key question is whether residential radon levels remain relatively stable over time. The authors should justify the temporal stability of these concentrations.

Response 1: Thank you for this important comment. A new explanatory paragraph was added to the Materials and Methods section explaining the scientific basis for treating indoor radon as temporally stable and therefore appropriate for long-term epidemiological analysis. In the revised manuscript this change can be found – page number 4 and lines 137-159.]

The national survey was conducted over a one-year period in two phases (summer and winter)…

Comments 2: The introduction could be better organized to more clearly articulate the background, rationale, and significance of the study.

Response 2: Thank you for the suggestion. A paragraph has been added to the introduction to further clarify the background and rationale. Beyond this addition, the introduction remains unaltered, as its current organization appropriately presents the necessary context. In the revised manuscript this change can be found – page number2 and lines 73-79.]

In Bulgaria, the reference levels for indoor radon concentration are defined…

Comments 3: Table 1 and Figure 1 appear to present the same content. It is suggested to retain only one of them. The Y-axis in Figure 1 should be clearly labeled, and the legend should be included directly in the figure.

Response 3:  I am grateful for this constructive comment. Although both Table 1 and Figure 1 have been kept to preserve their distinct visual contributions, the figure has been updated accordingly. The Y-axis is now explicitly labeled (‘Incidence’), and the legend (‘Incidence’, ‘Calendar year’) has been embedded directly within the figure for improved clarity. The changes made can be found in the revised manuscript– page number6, figure 1 and lines 213-217.]

Comments 4: The discussion section should also address other potential factors influencing lung carcinoma incidence. This would help contextualize the specific role of radon and highlight its significance relative to other risk factors. If a detailed discussion of these factors is beyond the scope of the current manuscript, the authors should at least acknowledge these potential confounders in the conclusion or the study outlook to provide a more balanced perspective.

Response 4: I appreciate your insightful comment, which has contributed to improving the quality of my work. A paragraph was added to the Discussion addressing the smoking prevalence variation (lack of regional data acknowledged), socioeconomic and ethnic factors and more.

It can be found – page number10 and lines 321 - 333.]

This study has several limitations. It is based on aggregated regional data…

Comments 5: The text in Lines 198–204 appears as a single-sentence paragraph and should be consolidated. The conclusion should be restructured into a continuous paragraph or a numbered list.

Response 5: I appreciate the reviewer’s observation. However, the content in Lines 198–204 (now lines 239-244) is intentionally formatted as a bullet point, and converting it into a paragraph would diminish its meaning and clarity. Thus, it has been retained in its original form. Discuss the changes made, providing the necessary explanation/clarification. By contrast, the Conclusions section has been restructured and expanded into continuous paragraphs in accordance with the journal’s formatting guidelines, with all essential points fully preserved. This change can be found – page number10, under Conclusions and lines 341-368.]

The average annual incidence of lung carcinoma in Bulgaria during the ten-year period…

Round 2

Reviewer 1 Report

Comments and Suggestions for Authors

Dear Author,

congratulation for the very good review work

Author Response

Thank you very much for your kind words and for recognizing the quality of the review work presented in the manuscript. Your positive evaluation and supportive feedback are truly appreciated.

Reviewer 2 Report

Comments and Suggestions for Authors

The paper can be accepted in the current form!

Author Response

I am grateful for your thorough assessment and for your encouraging remark that the paper can be accepted in its current form. Your review provides valuable validation of the clarity, rigor, and relevance of the study.